# Prevalence and predictors of kidney dysfunction among people living with HIV in Tanzania

Daniel Msilanga [1,2]* , Elizabeth Msangi [1,2]

**1** Department of Internal Medicine – Muhimbili University of Health and Allied Sciences, Dar es Salaam, Tanzania, **2** Department of Internal Medicine – Muhimbili National Hospital, Dar es Salaam, Tanzania

☉ This authors contributed equally to this manuscript
\* pascodanny07@gmail.com

## Abstract

As people living with HIV (PLHIV) in sub-Saharan Africa live longer due to widespread access to antiretroviral therapy (ART), the burden of non-communicable diseases, including kidney dysfunction (KD) has increased. Existing studies in Tanzania show varying prevalence and inconsistent predictors of KD, highlighting the need for updated, context-specific data. We conducted a cross-sectional sub-analysis of data from a larger study assessing point-of-care creatinine testing. PLHIV aged ≥18 years attending the HIV clinic at Temeke Regional Referral Hospital (TRRH) in Dar es Salaam from 5th January to 30th March 2025 consented to participate were included. Renal function was assessed using serum creatinine measured via the Jaffé method, and eGFR was calculated using the CKD-EPI 2021 equation. Kidney dysfunction was defined as eGFR < 60 mL/min/1.73 m². Logistic regression was used to identify predictors. Ethical approval was obtained from National Institute for Medical Research (NIMR) under reference number NIMR/HQ/R.8a/Vol.IX/4695. Among 358 participants, the majority were female (66.2%) and aged ≥45 years (62.3%). The prevalence of KD was 15.6% with 24.6% reporting at least one comorbid condition. In multivariable analysis, the presence of comorbidities was the only independent predictor of KD (aOR: 3.93; 95% CI: 1.85–8.36; p < 0.001). Only 5.3% of participants with reduced eGFR had a prior diagnosis of kidney disease. Kidney dysfunction is a significant but underdiagnosed comorbidity among PLHIV in urban Tanzania. Comorbid conditions, especially hypertension, are major contributors to reduced kidney function and integrating non-communicable disease screening and management into HIV care is needed to enable earlier detection and improve long-term renal outcomes.

## Introduction

With the scale-up of antiretroviral therapy (ART) in sub-Saharan Africa (SSA), people living with HIV (PLHIV) are living longer, with a change in the pattern of their illnesses from opportunistic infections to chronic non-communicable diseases such as cardiovascular,

**Data availability statement:** All relevant data are within the paper and its Supporting Information files.

**Funding:** The authors received no specific funding for this work.

**Competing interests:** The authors have declared that no competing interests exist.

liver, and kidney diseases [1–4]. Kidney dysfunction (KD) has emerged as a significant comorbidity in this population and is associated with increased morbidity and mortality and interruption of HIV care [5]. In SSA, the burden of KD among PLHIV is substantial ranging from 25% to 77% across different settings [6–9]. Tanzania is a clear example of this growing burden, with studies reporting a 15.7% to 20.7% prevalence of kidney dysfunction among PLHIV in rural areas, 25% to 32.8% among those initiating ART in urban areas [10–15]. These findings underscore that KD is an increasingly common health concern among PLHIV in the era of widespread ART coverage [1].

Although multiple Tanzanian studies have investigated KD in PLHIV, the predictors of dysfunction have varied widely [13,15]. Reported risk factors range from HIV-related factors, such as low CD4 count and high viral load, to traditional NCDs risk factors like older age, hypertension, obesity (BMI ≥ 30-kg/m$^2$), and female sex [11,15–17]. Additionally, ART-related variables such as regimen type and treatment duration have been implicated [18,19]. This variability may be due to differences in changing in burden of KD among PLHIV, population characteristics, healthcare setting (rural vs. urban), and definitions of KD used [11,18,19].

Given these discrepancies and evolving care patterns for PLHIV, along with shifting trends in kidney dysfunction (KD) across at-risk groups, there is a clear need for updated, context-specific data to inform screening and intervention strategies [11,15,16]. To address this gap, we conducted a study to determine the prevalence of kidney dysfunction and its predictors among PLHIV attending HIV clinic in Dar es Salaam, Tanzania. Our findings aim to complement existing evidence by providing localized insights into the risk factors influencing renal health in a contemporary urban HIV care setting.

## Methodology

### Ethics statement

The study received ethical approval from the National Institute for Medical Research (NIMR) under reference number NIMR/HQ/R.8a/Vol.IX/4695 and permission from Temeke Regional Referral Hospital management. All participants provided written informed consent (English and Swahili). Data were anonymized using unique identifiers, and confidentiality was maintained throughout. This sub-analysis adhered to the principles of the Declaration of Helsinki.

### Study design and setting

This study is a sub-analysis of data collected from a larger project titled *"Point-of-Care Creatinine Testing for Early Detection of Renal Dysfunction in Tanzanian HIV Patients."* While the parent study broadly assessed the diagnostic utility of point-of-care creatinine testing, this sub-analysis focuses specifically on estimating the prevalence and predictors of renal dysfunction among people living with HIV (PLHIV) [13]. We conducted a hospital-based cross-sectional study at the HIV clinic of Temeke Regional Referral Hospital (TRRH) in Dar es Salaam, Tanzania, over three months from the 5th of January to the 30th of March 2025. This urban public hospital provides antiretroviral therapy and chronic care services to a large population of PLHIV, making it an appropriate setting for evaluating kidney health in routine HIV care.

PLOS Global Public Health

### Recruitment procedures

We included PLHIV aged 18 years or older who were receiving care at TRRH. Participants were randomly selected from the clinic registry. Each patient was assigned a unique identification number from the clinic appointment book, and Probability Proportional to Size (PPS) sampling was used to select participants based on clinic volume. Recruitment continued until the target sample size was achieved.

All eligible patients were informed about the study, and those who provided written informed consent were enrolled. Data were collected using a structured, interviewer-administered electronic questionnaire, capturing sociodemographic and clinical information such as age, sex, ART history, CD4 count, viral load, and comorbidities. Blood pressure and body mass index (BMI) were measured and categorized according to WHO standards. Kidney function was assessed by measuring serum creatinine using the Jaffé method, with eGFR calculated using the 2021 CKD-EPI equation (non-race based), and KD was defined as eGFR of less than 60mls/min.

### Statistical analysis

Data were entered and analyzed using SPSS version 28. Continuous variables were summarized as means with standard deviations or medians with interquartile ranges, depending on data distribution. Categorical variables were presented as frequencies and percentages. The prevalence of renal dysfunction was calculated with corresponding 95% confidence intervals. Univariable logistic regression was performed to assess crude associations between potential predictors and renal dysfunction. All variables with a p-value <0.20 in univariate analysis, along with those of known clinical importance (e.g., age, sex), were included in the multivariate logistic regression model. Adjusted odds ratios with 95% confidence intervals were reported to identify independent predictors.

### Results

A total of 358 participants were included in this study. The median age was 48 years (IQR: 39–54), with most aged ≥45 years (62%). The majority were female (66.2%), married (57.8%), and had attained primary education (60.3%). Most participants were self-employed (66.5%), while 92.2% were non-smokers, and 65.4% reported former alcohol use (Table 1)

The median duration of HIV was 132 months (IQR: 84–180), with most participants on a Tenofovir Disoproxil Fumarate (TDF) based regimen (91.1%). A history of changing ART was reported by 19.8% of participants. Hypertension was the most common comorbidity (21.5%), and 24.6% had either hypertension, diabetes, or both. A total of 15.6% had reduced eGFR (<60 mL/min/1.73m²), while only 5.3% were told to have kidney disease. The mean systolic blood pressure was 123.2 mmHg (SD ± 17.7), and 58.4% were overweight or obese (BMI ≥ 25 kg/m²). Most participants (88.8%) had an undetectable HIV viral load (Table 2)

In the univariable analysis, older age (≥45 years) (OR: 3.25, 95% CI: 1.58–6.68, p = 0.001), presence of comorbidities (OR: 5.11, 95% CI: 2.81–9.29, p < 0.001), and detectable HIV viral load (OR: 2.80, 95% CI: 1.32–5.92, p = 0.007) were significantly associated with reduced eGFR. However, in the multivariable analysis, only the presence of comorbidities remained significant (aOR: 3.93, 95% CI: 1.85–8.36, p < 0.001) (Table 3).

### Discussion

Our study provides valuable insights into the burden and determinants of kidney dysfunction (KD) among people living with HIV (PLHIV) in an urban Tanzanian setting. We found that more than one in ten participants had KD, and nearly one-quarter reported comorbid conditions. In addition, the presence of comorbidities emerged as the only independent predictor of KD, underscoring the growing impact of non-communicable diseases on renal health in this population.

Hypertension was the most common comorbidity identified in our cohort, reflecting its high prevalence in both the general population and among HIV-positive individuals in Tanzania and across SSA as reported in other studies [20,21]. It is a well-established leading cause of kidney dysfunction and chronic kidney disease (CKD), and it is more

**Table 1. Socio-demographic of the study participants, n = 358.**

| Variable | Frequency (n) | Percent (%) |
|---|---|---|
| Age group (years) | | |
| < 45 | 135 | 37.7 |
| ≥ 45 | 223 | 62.3 |
| Median age in years (IQR) | 48 (39, 54) | |
| Gender | | |
| Male | 121 | 33.8 |
| Female | 237 | 66.2 |
| Marital status | | |
| Single | 65 | 18.2 |
| Married | 207 | 57.8 |
| Divorced | 46 | 12.8 |
| Widow | 40 | 11.2 |
| Level of education | | |
| No education | 11 | 3.1 |
| Primary | 216 | 60.3 |
| Secondary | 112 | 31.3 |
| Tertiary | 19 | 5.3 |
| Employment status | | |
| Unemployed (No employment) | 48 | 13.4 |
| Self employed | 238 | 66.5 |
| Employed | 63 | 17.6 |
| Retired | 9 | 2.5 |
| Cigarette smoking | | |
| Current smoker | 11 | 3.1 |
| Former smoker | 17 | 4.7 |
| Non smoker | 330 | 92.2 |
| Alcohol use | | |
| Current alcohol use | 80 | 22.3 |
| No alcohol use | 44 | 12.3 |
| Former alcohol use | 234 | 65.4 |

impactful when in the context of HIV [21,22]. The asymptomatic nature of hypertension often leads to late diagnosis and delayed treatment, which accelerates kidney damage and contributes to poor renal outcomes [23]. These findings underscore the critical importance of routine blood pressure screening and effective hypertension management within HIV care settings. Integrating non-communicable disease (NCD) services into HIV clinics may enable earlier detection and intervention for hypertension, thereby helping to mitigate one of the primary drivers of kidney dysfunction in both the general population and this particularly vulnerable group [1]

Mapesi et al. and Mwanjala et al. reported kidney dysfunction prevalence rates of 15.7% and 20.7%, respectively, among PLHIV in rural Tanzania, figures that align closely with our findings and highlight a substantial burden even among stable, ART-treated individuals [10,18]. In contrast, higher prevalence rates have been observed in urban and hospital-based settings, including 25% at ART initiation (Msango et al.) and 32.8% among patients on ART in referral centers [15,17]. These variations are likely due to the use of an eGFR cut-off of 90 mL/min/1.73 m², which reflects the definition of kidney dysfunction used [15]. Notably, only a small proportion of our participants with reduced eGFR had a prior diagnosis

**Table 2. Clinical characteristics of the study participants, n = 358.**

| Variable | Frequency (n) | Percent (%) |
|---|---|---|
| Median duration of HIV in months (IQR) | 132 (84, 180) | |
| Type of ART | | |
| TDF Based Regimen | 326 | 91.1 |
| Non TDF Based Regimen | 32 | 8.9 |
| History of changing ART | | |
| Yes | 71 | 19.8 |
| No | 287 | 80.2 |
| Comorbid conditions | | |
| Diabetes mellitus | 5 | 1.4 |
| Hypertension | 77 | 21.5 |
| Hypertension and Diabetes mellitus | 6 | 1.7 |
| Stroke | 2 | 0.6 |
| No comorbid condition | 268 | 74.9 |
| Standard EGFR (mL/min/1.73m²) | | |
| < 60 | 56 | 15.6 |
| ≥ 60 | 302 | 84.4 |
| Median standard EGFR (IQR) | 87.0 (68.8, 104.0) | |
| Known to have kidney diseases | | |
| Yes | 19 | 5.3 |
| No | 339 | 94.7 |
| Mean Systolic BP (±SD) | 123.2 (± 17.7) | |
| BMI category (kg/m²), | | |
| < 25 | 149 | 41.6 |
| ≥ 25 | 209 | 58.4 |
| HIV viral load | | |
| TND | 316 | 88.8 |
| Detectable | 40 | 11.2 |

Keys: TND: Target Not Detectable, ART: Anti-retroviral therapy, TDF: Tenofovir Disoproxil Fumarate

and knowledge, underscoring the problem of missed early stages of KD, which may be attributed to limited routine renal screening in HIV care programs and asymptomatic nature of the condition. These findings highlight the urgent need for accessible and affordable tools to facilitate early detection and timely intervention for kidney dysfunction within HIV services.

The presence of comorbid conditions was associated with nearly fourfold higher odds of reduced eGFR, with hypertension being the most common comorbid reported. These showcases the growing role of traditional non-communicable diseases (NCDs) in driving kidney dysfunction among PLHIV, alongside HIV-related risks. Similar findings have been reported in other Tanzanian and sub-Saharan African studies, where conditions like hypertension and obesity were strongly linked to impaired kidney function [18,19]. In contrast, studies among ART-naïve individuals have identified predictors more reflective of HIV disease severity, such as advanced WHO stage, high viral load, and low CD4 count [17]. These shifts suggest that as PLHIV live longer on ART, NCDs increasingly become dominant determinants of kidney health. This underscores the importance of integrating NCD care into HIV services to improve prevention and management of CKD in this population.

**Table 3. Univariable and multivariable analysis of the factors associated with the reduced eGFR.**

| Variable | Category | Univariable analysis | | | Multivariable analysis | | |
|---|---|---|---|---|---|---|---|
| | | cOR | 95% CI | p - value | aOR | 95% CI | p – value |
| Age (years) | ≥ 45 | 3.25 | 1.58 – 6.68 | 0.001 | 2.32 | 0.81 – 6.70 | 0.118 |
| | < 45 | Ref | | | | | |
| Gender | Male | 0.75 | 0.40 – 1.40 | 0.369 | | | |
| | Female | Ref | | | | | |
| Marital status | Single | 0.47 | 0.18 – 1.27 | 0.138 | 0.94 | 0.26 – 3.37 | 0.923 |
| | Divorced | 1.38 | 0.61 – 3.14 | 0.442 | 1.29 | 0.44 – 3.79 | 0.638 |
| | Widow | 2.15 | 0.98 – 4.76 | 0.058 | 1.08 | 0.39 – 2.97 | 0.888 |
| | Married | Ref | | | | | |
| Education | No education | 1.19 | 0.17 – 8.47 | 0.866 | | | |
| | Primary | 1.21 | 0.34 – 4.36 | 0.768 | | | |
| | Secondary | 0.58 | 0.15 – 2.31 | 0.441 | | | |
| | Tertiary | Ref | | | | | |
| Occupation | Unemployed | 1.76 | 0.71 – 4.36 | 0.225 | 1.57 | 0.45 – 5.43 | 0.476 |
| | Self employed | 0.58 | 0.27 – 1.25 | 0.164 | 0.77 | 0.27 – 2.21 | 0.624 |
| | Retired | 1.46 | 0.62 – 3.71 | 0.142 | 1.43 | 0.56 – 4.78 | 0.281 |
| | Employed | Ref | | | | | |
| Cigarette smoking | Smoker | 1.19 | 0.43 – 3.27 | 0.737 | | | |
| | Non smoker | Ref | | | | | |
| Alcohol use | Alcohol use | 0.79 | 0.43 – 1.47 | 0.464 | | | |
| | No alcohol use | Ref | | | | | |
| Comorbidities | Yes | 5.11 | 2.81 – 9.29 | < 0.001 | 3.93 | 1.85 – 8.36 | < 0.001 |
| | No | Ref | | | | | |
| BMI category | ≥ 25 | 1.12 | 0.63 – 2.01 | 0.700 | | | |
| | < 25 | Ref | | | | | |
| HIV viral load | Detectable | 2.80 | 1.32 – 5.92 | 0.007 | 2.23 | 0.78 – 6.39 | 0.137 |
| | TND | Ref | | | | | |

Key: cOR: crude Odds Ratio, aOR: adjusted Odds Ratio, Ref: Reference Category, TND: Target Not Detectable

Our study demonstrated a considerable burden of kidney dysfunction among people living with HIV in an urban Tanzanian clinic. Awareness of kidney dysfunction was very low, suggesting that routine care failed to identify at-risk patients despite regular healthcare contact. The strongest predictor of reduced kidney function was the presence of comorbid non-communicable conditions, while traditional HIV-related factors (e.g., immunosuppression or ART regimen) were not independent predictors. These findings show the urgent need to strengthen screening and early detection of kidney dysfunction in HIV care. Integrating NCD services, such as eGFR monitoring, blood pressure control, and diabetes management, into routine HIV care can facilitate earlier diagnosis and allow preventive measures, including optimizing blood pressure, avoiding nephrotoxic medications, and timely referral to specialist care, ultimately improving renal outcomes and slowing disease progression.

## Limitation

This study was a sub-analysis of data from a larger project focused on point-of-care creatinine testing, which may have influenced variable selection and introduced bias, although key predictors of kidney dysfunction were included. Conducted at a single urban HIV clinic in Dar es Salaam, the findings may not be generalizable to rural settings or other regions with

different patient profiles and healthcare access, and there is potential for selection bias as only individuals engaged in care were enrolled. Additionally, renal function was assessed using a single creatinine measurement without confirmatory testing, limiting our ability to distinguish chronic kidney disease from transient or acute changes, such as dehydration or acute kidney injury. Despite these limitations, the study offers valuable insights into the burden and predictors of kidney dysfunction in a real-world HIV care setting and highlights key risk factors relevant for improving patient management. Moreover, proteinuria was not assessed using urinary albumin-to-creatinine ratio (UACR), which may have resulted in missed identification of individuals with early-stage CKD and preserved eGFR. The absence of this marker likely led to underestimation of the true CKD burden. Despite these limitations, the study provides valuable insights into the prevalence and predictors of kidney dysfunction in a real-world HIV care setting and highlights key risk factors relevant for improving patient management.

## Acknowledgments

We gratefully acknowledge the support of Temeke Regional Referral Hospital for granting approval to conduct this research. We also extend our sincere appreciation to the patients who participated in the study.

## Author contributions

**Conceptualization:** Daniel Msilanga, Elizabeth Msangi.

**Data curation:** Daniel Msilanga, Elizabeth Msangi.

**Formal analysis:** Daniel Msilanga, Elizabeth Msangi.

**Investigation:** Daniel Msilanga.

**Methodology:** Daniel Msilanga, Elizabeth Msangi.

**Project administration:** Daniel Msilanga.

**Resources:** Daniel Msilanga.

**Supervision:** Daniel Msilanga.

**Validation:** Elizabeth Msangi.

**Writing – original draft:** Daniel Msilanga, Elizabeth Msangi.

**Writing – review & editing:** Daniel Msilanga, Elizabeth Msangi.

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
