## [Decision Letter · Decision Letter 0]

PGPH-D-25-00958

Prevalence and Predictors of Kidney Dysfunction among People Living with HIV in Tanzania

Dear Dr. Msilanga,

Thank you for submitting your manuscript to PLOS Global Public Health. After careful consideration, we feel that it has merit but does not fully meet PLOS Global Public Health’s publication criteria as it currently stands. Therefore, we invite you to submit a revised version of the manuscript that addresses the points raised during the review process.

We look forward to receiving your revised manuscript.

Kind regards,

Elliot Koranteng Tannor, MBChB, FWACP, MPhil(Neph), Cert Neph(SA), MBA

Academic Editor

Journal Requirements:

Additional Editor Comments (if provided):

Reviewers' comments:

Reviewer's Responses to Questions

**Comments to the Author**

1. Does this manuscript meet PLOS Global Public Health’s publication criteria?

Reviewer #1: Yes

Reviewer #2: Yes

2. Has the statistical analysis been performed appropriately and rigorously?

Reviewer #1: Yes

Reviewer #2: Yes

3. Have the authors made all data underlying the findings in their manuscript fully available (please refer to the Data Availability Statement at the start of the manuscript PDF file)?

Reviewer #1: Yes

Reviewer #2: Yes

4. Is the manuscript presented in an intelligible fashion and written in standard English?

Reviewer #1: Yes

Reviewer #2: Yes

Reviewer #1: A simple but informative study which was well-written. kindly correct the following observations.

Manuscript Number: PGPH-D-25-00958

Prevalence and Predictors of Kidney Dysfunction among People Living with HIV in Tanzania

Lines 71-72: Kindly reference the statement, “This study is a sub-analysis of data collected from a larger project titled “Point-of-Care Creatinine Testing for Early Detection of Renal Dysfunction in Tanzanian HIV Patients.”

Lines 140-141: Provide explanation why the prevalence of kidney dysfunction of 15.6% among PLHIV, in urban setting is similar to 15.7% and 20.7% found in rural hospital-based setting. Is there a similarity in the patients profile, standard of healthcare system and use of ART?

Lines 144-145: Following the statement, “These variations likely reflect differences in patient characteristics, healthcare levels, and definitions of kidney dysfunction (15)”, kindly provide possible reasons why this study found the prevalence of kidney dysfunction of 15.6% among PLHIV, and others that found higher prevalences of 25% and 32.8% in similar urban hospital-based settings.

Reviewer #2: Proteinuria (UACR) would have assisted in identifying patients with CKD and normal creatinine (eGFR). This can be added to the limitations; if this data is available, it should be added to the manuscript.

**Do you want your identity to be public for this peer review?** For information about this choice, including consent withdrawal, please see our Privacy Policy

Reviewer #1: No

Reviewer #2: No

---

## [Decision Letter · Decision Letter 1]

PGPH-D-25-00958R1

Prevalence and Predictors of Kidney Dysfunction among People Living with HIV in Tanzania

Dear Dr. Msilanga,

Thank you for submitting your manuscript to PLOS Global Public Health. After careful consideration, we feel that it has merit but does not fully meet PLOS Global Public Health’s publication criteria as it currently stands. Therefore, we invite you to submit a revised version of the manuscript that addresses the points raised during the review process.

We look forward to receiving your revised manuscript.

Kind regards,

Elliot Koranteng Tannor, MBChB, FWACP, MPhil(Neph), Cert Neph(SA), MBA

Academic Editor

Journal Requirements:

Additional Editor Comments (if provided):

Reviewers' comments:

Reviewer's Responses to Questions

**Comments to the Author**

Reviewer #1: All comments have been addressed

Reviewer #3: (No Response)

publication criteria?

Reviewer #1: Yes

Reviewer #3: Yes

3. Has the statistical analysis been performed appropriately and rigorously?

Reviewer #1: Yes

Reviewer #3: Yes

4. Have the authors made all data underlying the findings in their manuscript fully available (please refer to the Data Availability Statement at the start of the manuscript PDF file)?

Reviewer #1: Yes

Reviewer #3: Yes

5. Is the manuscript presented in an intelligible fashion and written in standard English?

Reviewer #1: Yes

Reviewer #3: Yes

Reviewer #1: The author has attached a table of the point by point response to the comments, but, the correction is not reflected in the attached manuscript. The manuscript attached was the old version.

If the corrections are reflected in the manuscript, then I recommend approval for publication.

Reviewer #3: The authors have responded appropriately to the initial review comments however a few minor comments need to be addressed.

Line 98 and Table 2: Unit of eGFR should be appropriately captured as milliliters per minute per 1.73 square meters (mL/min/1.73m²) throughout the manuscript.

Line 105, 106 and Table 3: Univariate and multivariate is used interchangeably with univariable and multivariable throughout the manuscript. Kindly stick to the use of univariable and multivariable.

Table 2: Kindly stick to the use of the abbreviation “eGFR” throughout the manuscript.

Table 3: “No - Ref” under Alcohol use and Comorbidities should be removed.

References: Ref 3 is incomplete

**Do you want your identity to be public for this peer review?** For information about this choice, including consent withdrawal, please see our Privacy Policy

Reviewer #1: No

Reviewer #3: No

---

## [Decision Letter · Decision Letter 2]

Prevalence and Predictors of Kidney Dysfunction among People Living with HIV in Tanzania

PGPH-D-25-00958R2

Dear Dr Msilanga,

We are pleased to inform you that your manuscript 'Prevalence and Predictors of Kidney Dysfunction among People Living with HIV in Tanzania' has been provisionally accepted for publication in PLOS Global Public Health.

Best regards,

Elliot Koranteng Tannor, MBChB, FWACP, MPhil(Neph), Cert Neph(SA), MBA

Academic Editor

Reviewer Comments (if any, and for reference):

Reviewer's Responses to Questions

**Comments to the Author**

Reviewer #3: All comments have been addressed

publication criteria?

Reviewer #3: Yes

3. Has the statistical analysis been performed appropriately and rigorously?

Reviewer #3: Yes

4. Have the authors made all data underlying the findings in their manuscript fully available (please refer to the Data Availability Statement at the start of the manuscript PDF file)?

Reviewer #3: Yes

5. Is the manuscript presented in an intelligible fashion and written in standard English?

Reviewer #3: Yes

Reviewer #3: Minor edits:

1. Title page (Line 18): *This authors contributed equally to this manuscript. Correct to "These authors..."

2. eGFR should be written out in full form at first use.

**Do you want your identity to be public for this peer review?** For information about this choice, including consent withdrawal, please see our Privacy Policy

Reviewer #3: No
